# Longitudinal Fibular Deficiency: A Cross-Sectional Study Comparing Lower Limb Function of Children and Young People with That of Unaffected Peers

**DOI:** 10.3390/children6030045

**Published:** 2019-03-15

**Authors:** Joshua W. Pate, Mark J. Hancock, Louise Tofts, Adrienne Epps, Jennifer N. Baldwin, Marnee J. McKay, Joshua Burns, Eleanor Morris, Verity Pacey

**Affiliations:** 1Faculty of Medicine and Health Sciences, Macquarie University, Sydney, New South Wales 2109, Australia; mark.hancock@mq.edu.au (M.J.H.); eleanor.morris@health.nsw.gov.au (E.M.); verity.pacey@mq.edu.au (V.P.); 2The Children’s Hospital at Westmead, Westmead, New South Wales 2145, Australia; louise.tofts@health.nsw.gov.au (L.T.); adrienne.epps@health.nsw.gov.au (A.E.); joshua.burns@sydney.edu.au (J.B.); 3Faculty of Health Sciences, The University of Sydney, Sydney, New South Wales 2006, Australia; jennifer.baldwin@aut.ac.nz (J.N.B.); marnee.mckay@sydney.edu.au (M.J.M.); 4School of Clinical Sciences, Faculty of Health and Environmental Sciences, Auckland University of Technology, Auckland 1142, New Zealand

**Keywords:** longitudinal fibular deficiency, lower limb function, children, young people, unaffected peers, KOOS, KOOS-Child, CAIT, CAIT-Youth

## Abstract

Longitudinal fibular deficiency (LFD), or fibular hemimelia, is congenital partial or complete absence of the fibula. We aimed to compare the lower limb function of children and young people with LFD to that of unaffected peers. A cross-sectional study of Australian children and young people with LFD, and of unaffected peers, was undertaken. Twenty-three (12 males) children and young people with LFD (74% of those eligible) and 213 unaffected peers, all aged 7–21 years were subject to the Knee Osteoarthritis Outcome Score (KOOS/KOOS-Child) and the Cumberland Ankle Instability Tool (CAIT/CAIT-Youth). Linear regression models compared affected children and young people to unaffected peers. Participants with LFD scored lower in both outcomes (adjusted *p* < 0.05). The difference between participants with LFD and unaffected peers was significantly greater among younger participants than older participants for KOOS activities and sports domain scores (adjusted *p* ≤ 0.01). Differences in the other KOOS domains (pain/symptoms/quality of life) and ankle function (CAIT scores) were not affected by age (adjusted *p* ≥ 0.08). Children and young people with LFD on average report reduced lower limb function compared to unaffected peers. Knee-related activities and sports domains appear to be worse in younger children with LFD, and scores in these domains become closer to those of unaffected peers as they become older.

## 1. Introduction

Longitudinal fibular deficiency (LFD), also known as fibular hemimelia, is the congenital partial or complete failure of formation of the fibula [1]. It is not an isolated anomaly but a spectrum of dysplasia of the lower limb [2]. LFD is the most common long-bone deficiency [3], with an estimated incidence of 7.4–20 cases per million live births [4,5].

LFD may present unilaterally or bilaterally, with varying severity and involvement of associated anatomical changes [6]. The integrity and function of both the knee and ankle joint may be affected, as well as the length of the long bones and therefore the leg. As the fibula normally contributes to the structure and stability of the lateral part of the ankle joint, its absence in conjunction with absent or insufficient ligaments in the ankle and knee, particularly the anterior cruciate ligament, can significantly affect joint stability and therefore may affect lower limb function and participation in daily activities [7]. It is commonly associated with an equinovalgus [2,3,6,8,9,10,11,12,13] or equinovarus [3,6,9,10] foot, in addition to lateral ray deficiencies, tarsal coalition, anteromedial bowing of the tibia, genu valgum, hypoplastic patella, cruciate ligament deficiency [9,13], and femoral shortening.

Parents of children and young people with any congenital structural anomaly are concerned about their child’s ability to perform activities of daily living and fully participate in society throughout their life [14]. To date, studies on LFD have focused on impairments rather than the domains of activity limitations and participation restrictions, as described by the International Classification of Functioning Children & Youth Version (ICF-CY) [15]. Commonly used outcome measures in the literature, such as healing index, limb length, and joint range of motion [2,3,11], focus on impairments but do not include the impact that these impairments have on a young person’s lower limb function. Patient-reported outcomes are likely to better capture the impact LFD has on function and quality of life.

Adults with LFD have been reported to have similar lower limb function to that of their unaffected peers in three studies [16,17,18]. No published information is available for lower limb function in children and young people with LFD despite these being the life stages where functional skills and social relationships develop. Therefore, the aim of this cross-sectional study was to investigate patient-reported lower limb function on the affected limb in children and young people with LFD in comparison to unaffected peers.

## 2. Materials and Methods

### 2.1. Study Design

This was a cross-sectional study involving children and young people with LFD who lived in the state of New South Wales, Australia, from October 2015 to October 2017. Measures of lower limb function were collected and compared to a data subset from the 1000 Norms Project, a cross-sectional study that collected patient-reported outcomes and physical performance measures in 1000 healthy individuals aged 3–101 years (January 2014–September 2015), also living in New South Wales, Australia [19]. Ethics approval was gained for the study from Sydney Children’s Hospitals Network (LNR/15/SCHN/327) and Macquarie University (Ref: 5201500761). The 1000 Norms Project had ethical approval from the institutional ethics committee at the University of Sydney (HREC 2013/640).

### 2.2. Participants

An attempt was made to identify the complete population of individuals aged 7–21 years of age, with a diagnosis of LFD, who were living in New South Wales during the study period (October 2015–October 2017). Exclusion criteria included any individuals with unassociated comorbidities likely to significantly affect lower limb function and quality of life, such as those with an intellectual disability or a neurological condition, those who had undergone previous lower limb joint surgery not related to their condition, and those having undergone a surgical procedure related to LFD within the previous 6 months. Based on an estimated incidence of between 7.4 and 20 cases per million live births [4,5], and the 2014 New South Wales Census population data [20], the number of possible participants with LFD in the study’s age range was estimated to be between 10 and 28. Each participant’s condition was classified using the Achterman & Kalamchi (1979) system in which hypoplasia of the fibula is Type Ia, partial absence is Type Ib, and complete absence is Type II [9].

The control group was comprised of a sample of individuals living in New South Wales, Australia, who were representative of the healthy ‘normal’ population across this age range [19]. Participants in this study were healthy by self-report and able to participate in age-appropriate daily activities. Potential participants in the 1000 Norms Project were excluded based on an inability to follow instructions, insufficient English language proficiency, and any condition affecting neurological function and mobility.

### 2.3. Identification and Recruitment of Participants with LFD and Unaffected Peers

Participants with LFD were identified via two sources: (1) through diagnostic codes in the rehabilitation and orthopaedic databases at the Sydney Children’s Hospitals Network, which provides the only paediatric management clinics for children and young people with LFD in the state of New South Wales, and (2) through a patient support organization called ‘Limbs4Life’. Potential participants and their parent/carer were sent a letter of invitation, participant information sheet, three questionnaires, and a stamped self-addressed envelope. Potential participants who did not respond within two weeks received a follow-up telephone call, email, or text message. Three attempts were made to contact potential participants who did not reply to the first letter of invitation. All participants in the 1000 Norms Project that were aged 7–21 years were included as the control group of unaffected peers.

### 2.4. Data Collected

After obtaining informed consent from participants, questionnaire and demographic data were collected by mail or email depending on participant preference. Demographic data were collected from the parent/carer for children (ages 7–16 years) and from the participant for young people (ages 17–21 years) using a standard questionnaire. Data collected included age, gender, affected leg(s), height, weight, prosthetic use, pain, falls history, and surgical history. Treating health professionals confirmed the patient-reported surgical procedures and dates these occurred.

Control data for unaffected peers were available at an individual participant level for young people aged 7–21 years of age who participated in the 1000 Norms Project [19].

### 2.5. Outcomes

Patient-reported knee function was assessed through the Knee Osteoarthritis Outcome Score (KOOS/KOOS-Child). The KOOS was used for participants aged 17–21 years, and the paediatric version of the KOOS, the KOOS-Child, was used for participants aged 7–16 years. The KOOS and KOOS-Child are patient-reported outcome measures used to assess an individual’s knee function. The KOOS-Child mirrors the KOOS, and both questionnaires have five domains: pain, other symptoms, function in daily living, function in sport and recreation, and knee-related quality of life. The KOOS/KOOS-Child are valid and reliable outcome measures [21,22] with a focus on the activity limitation and participation restriction components of the ICF-CY [15]. A score from 0 to 100 is given for each of the domains, a higher score indicating better function.

To investigate functional ankle instability, the Cumberland Ankle Instability Tool (CAIT/CAIT-Youth) was used—the CAIT for participants aged 17–21 years and CAIT-Youth for participants aged 7–16 years. It contains nine questions with a total score out of 30, a higher score indicating better function. The CAIT and CAIT-Youth are valid and reliable patient-reported outcome measures [23,24], and questions relate to ankle pain and functional instability in a variety of environmental contexts including sports participation. Children and young people who had previously undergone a through-ankle amputation did not complete this questionnaire.

If participants did not complete all questions in the KOOS or CAIT, each domain with missing data was scored following the standard instructions provided in each questionnaire. The most affected limb was used in all analyses for participants who had bilateral limb involvement. The control participants from the 1000 Norms Project completed the same outcomes as the participants with LFD.

An open-ended question was asked to identify the challenges of growing up with LFD from the perspective of the patients and/or their parents/carers. All affected participants were asked, ‘Have you [or has your child] faced challenges growing up where you could have been assisted to prepare and deal with these challenges? If ‘Yes’, please list the top 3 (in priority order)’. Common themes raised in these responses were identified and the frequency of responses relating to identified themes was recorded.

### 2.6. Statistical Analysis

A descriptive analysis of the demographics of the individuals with LFD and control participants was performed. As the paediatric versions of the KOOS and CAIT, the KOOS-Child and CAIT-Youth, respectively, are similar to the adult versions but with simplified language to make them appropriate for children, a combined analysis was performed for the two versions of each outcome. Individual participant data for both children and young people with LFD and the unaffected peers were combined in a single data set. Comparison of KOOS and CAIT scores for children and young people with LFD and unaffected peers was performed using linear regression models. Separate models were built for each outcome including the five KOOS domains and a total CAIT score. For each outcome a simple model was built where LFD status (‘yes’ or ‘no’) was the only variable entered. Next a model was built adjusting for age, gender, and body mass index-for-age percentile [25]. Finally, a model was built adjusting for the same variables, but it also included the interaction between age and LFD. The interaction term was included to assess whether differences in outcomes between individuals with LFD and unaffected peers were systematically related to age. The data were assessed for normality by visual inspection, and means (SD) or medians (IQR) were used based on this assessment. All analyses were conducted using Statistical Package for the Social Sciences, version 22.0 [26].

## 3. Results

A total of 31 potential participants were initially identified through the clinic database and an attempt to contact them was made. Two potential participants contacted the researchers through Limbs4Life, but both were too young to participate in the study. Two potential participants declined to participate, and six did not respond to contact attempts. Therefore, 23 children and young people (74%) agreed to participate in the study, and all completed the questionnaires. Normative data were obtained from 213 unaffected peers who were in the appropriate age range from the 1000 Norms study [19].

Details of included participants are provided in Table 1. Within the group of individuals with LFD (*n* = 23), 12 participants were affected on the right side, 6 participants were affected on the left, and the remaining 5 participants had bilateral involvement. Total fibula absence was identified in two participants. The number and type of interventions that participants underwent varied, with most participants having multiple procedures; seven participants had Syme’s amputations, seven had lengthening procedures, eight underwent epiphysiodesis, and five participants had been managed conservatively. Five children with LFD, all under the age of 15 years, reported 1–3 falls in the past week.

The KOOS/KOOS-Child questionnaire was fully completed by 22 of 23 participants with LFD, and partially completed by one participant who completed only two of the KOOS domains. This questionnaire was completed by 211 of 213 participating unaffected peers, as there were 2 cases with missing data.

The CAIT/CAIT-Youth was completed by 19 of 23 participants, as there were 4 participants who had missing data and did not respond to requests to complete these questions. This questionnaire was completed by 208 of 213 participating unaffected peers, as there were 5 cases with missing data.

A comparison with unaffected peers of age, gender, and body mass index for age percentile is provided in Table 2. There were no significant differences between groups for age (*p* = 0.81), gender (*p* = 0.72), or BMI-for-age-percentile (*p* = 0.32).

The outcome scores for individuals with LFD and unaffected peers are presented in Table 3 along with unadjusted and adjusted differences between the two groups. There was a statistically significant difference between participants with LFD and unaffected peers for all KOOS domains and the CAIT score. The between-group differences from the unadjusted model were similar to the between-group differences from the adjusted model. Adjusted between-group differences ranged from 7.2 points for the KOOS activities domain to 23.2 points for the KOOS quality of life domain. Unaffected peers scored higher in all outcomes.

For KOOS activities and sport domains, age significantly affected the difference in scores between participants with LFD and unaffected peers (Figure 1, *p* ≤ 0.006 for age interactions). The differences between participants with LFD and unaffected peers were greatest for the younger children and approximated normal values in the older children. As the age of participants increased, the difference between KOOS scores for participants with LFD and unaffected peers, for each successive year, reduced by 1.1 (95% CI: 0.3–1.9) points for activities of daily living and by 2.0 (95% CI: 0.6–3.3) points for sports and recreation. For the other three KOOS domains (pain/symptoms/quality of life), age did not significantly affect the difference in scores (Figure 1, *p* ≥ 0.085 for age interactions). The differences in CAIT scores between participants with LFD and unaffected peers were also not influenced by age (Figure 2; *p* = 0.256 for age interaction).

Twelve out of 23 participants with LFD reported challenges in the open-ended question. The most frequent responses related to anxiety (5 mentions), social acceptance (4 mentions), and sports participation (4 mentions).

## 4. Discussion

This study found that children and young people with LFD on average report reduced lower limb function when compared to unaffected peers. A novel finding from this study was that knee-related activities and sports domains for people with LFD are significantly worse than unaffected children, but similar to those for unaffected young people. In contrast, ankle function and knee-related pain, symptoms, and quality of life are reduced in both children and young people with LFD when compared to unaffected peers.

The findings of this study, that children and young people with LFD have reduced knee and ankle function compared to unaffected peers, supplement the previous literature regarding adults. Adults with LFD had active lives similar to age-matched controls and high levels of general function using the Short Form 36 (SF-36) [17,18]. Likewise, Birch et al. (1999) investigated the quality of life in two adult age groups with LFD and found it similar to that of the unaffected adult population [16]. For the younger participants in our study, and for ankle function specifically, there is no previous literature to use for comparison. The novel findings in this younger age group warrant further investigation.

The finding that younger children with LFD are substantially behind unaffected peers with regard to self-reported knee function suggests that early intervention targeting this outcome may be important to meet their functional potential in the short term and ‘catch up’ with their unaffected peers sooner. Guidelines state that comprehensive multidisciplinary services incorporating both physical and psychological management are likely to improve the outcomes for children and young people with LFD [27]. While current practice in Australia appears to broadly adhere to these guidelines, the majority of affected children and young people do not receive interventions such as anterior cruciate ligament (ACL) ligamentoplasty and physiotherapy exercises specifically aiming to improve knee function. Longitudinal studies following individual children and young people over time are needed to further investigate this study’s finding that some aspects of knee function of affected children appear to more closely resemble those of unaffected peers as they grow older.

Ankle function scores were lower for children and young people with LFD in comparison to unaffected peers regardless of age. Despite the nature of the anatomical consequences that may impact ankle instability, there are currently no studies providing evidence on strategies for improving ankle function, such as physiotherapy and prosthetics, in individuals with this condition. The benefits of physiotherapy to improve sensorimotor function have been well established in unaffected young adults with chronic functional ankle instability [28]. Physiotherapy aimed at improving the functional ankle stability of individuals with LFD may be important at any age, and this should be investigated.

A strength of this study was the large control group of unaffected peers who were similar in terms of the baseline characteristics and time period when data were collected. By having the same patient level data for both cases and controls, adjusted between-group differences could be calculated using modelling that included potential confounders. This study was also able to investigate interactions between age and LFD status. Although this study only had 23 participants with LFD, this represents a large proportion of the whole known population (74%) in the state of NSW, Australia. Estimates of the LFD population from the census data and incidence rates suggest most, if not all, potential participants are likely to have been identified.

This study has several limitations. While this study did adjust for some variables, it was not possible to adjust for other factors such as different severities of disease and treatments provided, which could have impacted the findings. To achieve an adequate sample size to compare between the different severity classifications of LFD, large multi-country studies would need to be undertaken. In addition, no condition-specific quality of life measure currently exists. Although the KOOS and CAIT have been widely used in a variety of different patient populations, these measures have not been specifically validated in assessing individuals with LFD. To inform the development of such a measure, the main challenges presented by the participants of this study could be used as a basis for more qualitative work.

Important areas for future research include investigations to better understand the factors contributing to the range of functional levels in children and young people with LFD. While we found on average that function was below that of unaffected peers, some individuals scored at or above the average levels of unaffected peers. Large future studies comparing disease severities, or studies able to account for the individual’s severity of LFD, may delineate why some children and young people have reduced function and others do not, and provide suggestions that lead to better functional outcomes in children and young people with LFD. The responses to the open-ended question highlight the difficulties with anxiety and social acceptance for families of children with LFD, a broader area where further research is required [29]. It is also important to consider if patient-reported outcomes are related to specific objective measures that provide guidance on treatment options.

The results of this study provide helpful and relevant clinical information to families and health professionals regarding the functional outcomes of young people with LFD and how this varies across different ages. Detailed information about the LFD population from a more holistic perspective encompassing all aspects of the ICF-CY may enable improved education. The provision of this information may reduce anxiety for families, particularly in the early stages of the child’s life regarding the likely functional impact of the condition, and it could also assist families in making better long-term management decisions.

## 5. Conclusions

Children and young people with LFD on average reported reduced lower limb function compared to unaffected peers. Knee-related activities and sports domains appeared to be worse in children with LFD but closer to values of unaffected young people. This study provides helpful information about function, which is important for patients and their families. This study also identifies opportunities for improving clinical care and directing future research.

## Figures and Tables

**Figure 1 children-06-00045-f001:**
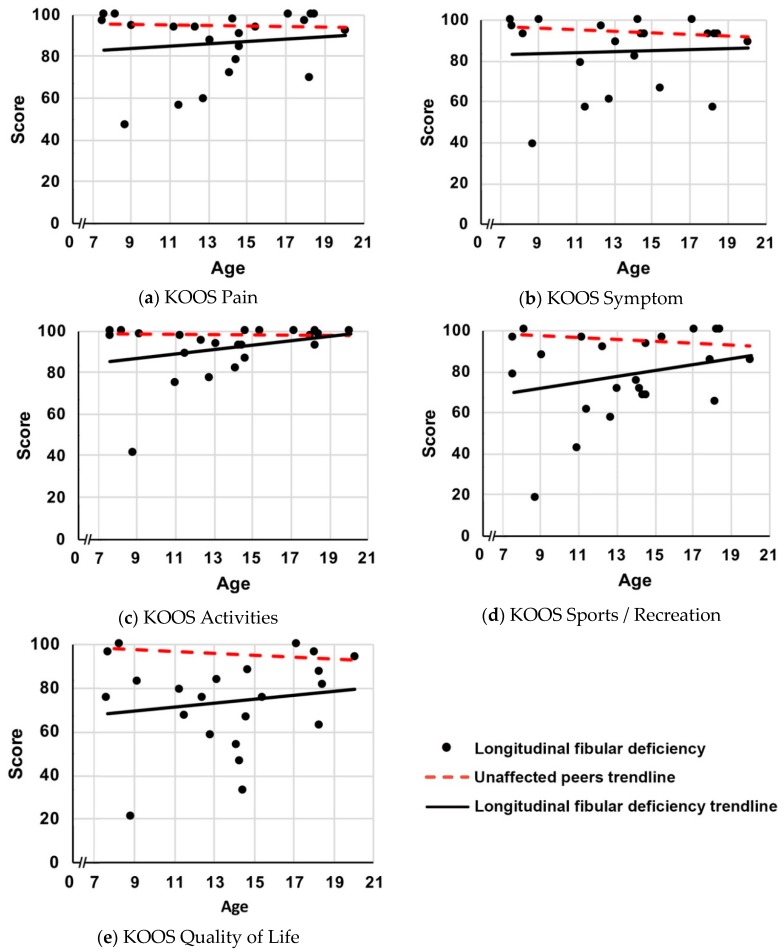
Scores of children and young people with LFD for intact knees on the affected limb and trendline, and the unaffected peers trendline for each KOOS/KOOS-Child domain against age (years): (**a**) KOOS Pain; (**b**) KOOS Symptoms; (**c**) KOOS Activities; (**d**) KOOS Sports/Recreation; (**e**) KOOS Quality of Life.

**Figure 2 children-06-00045-f002:**
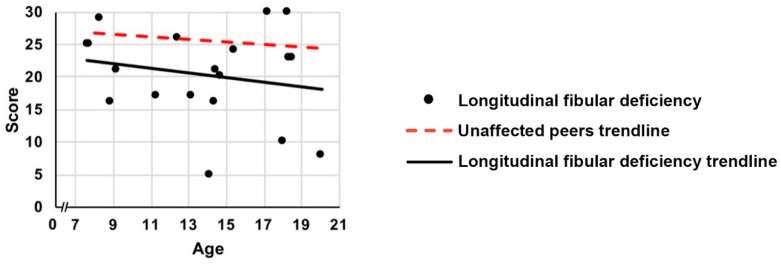
Scores of children and young people with LFD for intact ankles on the affected limb and trendline, and the unaffected peers trendline for the CAIT/CAIT-Y against age (years).

**Table 1 children-06-00045-t001:** Demographics of the participants with longitudinal fibular deficiency (LFD).

Characteristic	LFD (*n* = 23)
Bilateral (*n*, %)	5/23 (22%)
Unilateral side affected (right, %)	12/18 (67%)
Classification of every affected limb ^1^ (IA, IB, II)	23, 3, 2
Number of foot rays (Median, IQR)	4 (3.5–5)
Number of orthopaedic procedures (median, IQR)	1.5 (1–2)
Amputation (*n*, %, age range in years)	8 ^2^ (35%), 8–18
Leg lengthening (*n*, %, age range in years)	7 (30%), 11–18
Epiphysiodesis (*n*, %, age range in years)	8 (35%), 11–20
Number of falls in past week (Median, IQR)	0 (0–1)

^1^ Participants affected limbs were classified using the Achterman and Kalamchi (1979) system as Type IA, IB, or II [9]. The values given here are the number of limbs. Type IA—fibula hypoplastic but whole; Type IB—part of fibula absent; Type II—true agenesis of the fibula. ^2^ Seven participants had a Syme’s amputation, and one participant had toes amputated.

**Table 2 children-06-00045-t002:** Comparison between the children with LFD and unaffected peers.

Characteristic	Children with LFD (*n* = 23)	Unaffected Peers (*n* = 213)
Age in years (Mean, SD, range)	13.5 (3.8), 7–20	13.7 (3.8) 8–20
Gender (Female, %)	11/23 (48%)	110/213 (52%)
Body Mass Index For Age Percentile (Mean, SD, range)	54 (28.6), 5.9–97.7	60 (27.0), 2.3–98.8

**Table 3 children-06-00045-t003:** KOOS and CAIT outcomes for the LFD group and the unaffected peers group.

	Outcome	LFD	Unaffected Peers	Unadjusted Difference between Groups (95% CI)	*p* value	Adjusted ^3^ Difference between Groups (95% CI)	*p* Value (Adjusted Difference)
*Knee function* ^1^	***Pain (/100)***	*n* = 22	*n* = 211				
	*Mean score (SD)*	86.4 (15.9)	94.9 (11.4)	−8.5 (−13.8 to −3.3)	0.002	−8.8 (−14.2 to −3.4)	0.001
	***Symptoms (/100)***	*n* = 22	*n* = 211				
	*Mean score (SD)*	84.7 (17.3)	94.4 (9.2)	−9.6 (−14.2 to −5.1)	<0.001	−9.6 (−14.1 to −5.0)	<0.001
	***Activities of Daily Living (/100)***	*n* = 23	*n* = 211				
	*Mean score (SD)*	91.6 (13.3)	98.2 (5.4)	−6.6 (−9.4 to −3.7)	<0.001	−7.2 (−10.1 to −4.3)	<0.001
	***Sports/Recreation (/100)***	*n* = 23	*n* = 211				
	*Mean score (SD)*	78.6 (21.0)	95.6 (10.3)	−16.9 (−22.0 to −11.9)	<0.001	−17.9 (−23.1 to −12.8)	<0.001
	***Quality of life (/100)***	*n* = 22	*n* = 211				
	*Mean score (SD)*	73.7 (21.3)	95.6 (10.7)	−22.0 (−27.3 to −16.7)	<0.001	−23.2 (−28.8 to −17.7)	<0.001
*Ankle function* ^2^	***Total (/30)***	*n* = 19	*n* = 208				
	*Mean score (SD)*	20.3 (7.1)	25.6 (4.8)	−5.3 (−7.7 to −2.9)	<0.001	−6.6 (−9.0 to −4.3)	<0.001

^1^ Knee function measured with the Knee Osteoarthritis Outcome Score (KOOS/KOOS-Child)—higher scores indicate better function. ^2^ Ankle function measured with the Cumberland Ankle Instability Tool (CAIT/CAIT-Youth)—higher scores indicate better function. ^3^ Adjusted for age, gender, and body-mass-index-for-age percentile.

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
