# Peer review of "Longitudinal Fibular Deficiency: A Cross-Sectional Study Comparing Lower Limb Function of Children and Young People with That of Unaffected Peers"

_children, 2019, doi:10.3390/children6030045_

Round 1

Reviewer 1 Report

Thank you for your paper "Longitudinal Fibular Deficiency: A cross-sectional study comparing lower limb function of children and young people with unaffected peers". This study compared the KOOS/Child and CAIT outcome scores with those of unaffected peers in the same geographic area. The authors should be commended for this importants work looking at the affect of fibular hemimelia on children and patient/parent reported outcomes. This study answers valuable questions and identifies areas for future research and interventions that is unique.

One suggestion is to clarify the last sentence in the abstract (line 28). The statement "...appear to be worse in younger children with LFD and closer to healthy values in young people." is confusing

Author Response

PDF response to Reviewer 1's comments - attached.

Reviewer 2 Report

This is an interesting study and I raccomand the pubblication with minor revision. This study compares fibular hemimelia patient with normal population. However, this has the great limitation to not compare severe vs mild fibular hemimelia. I would raccomand to report in the text has limitation and propose it as a second part of this work.

Author Response

PDF response to Reviewer 2's comments - attached.
